# Scaling Memory-Augmented Neural Networks with Sparse Reads and Writes

**Jack W Rae**[*]  **Jonathan J Hunt**[*]  **Tim Harley**  **Ivo Danihelka**  **Andrew Senior**
jwrae            jjhunt             tharley        danihelka        andrewsenior

**Greg Wayne**          **Alex Graves**          **Timothy P Lillicrap**
gregwayne              gravesa                  countzero

Google DeepMind
@google.com

## Abstract

Neural networks augmented with external memory have the ability to learn algorithmic solutions to complex tasks. These models appear promising for applications such as language modeling and machine translation. However, they scale poorly in both space and time as the amount of memory grows — limiting their applicability to real-world domains. Here, we present an end-to-end differentiable memory access scheme, which we call Sparse Access Memory (SAM), that retains the representational power of the original approaches whilst training efficiently with very large memories. We show that SAM achieves asymptotic lower bounds in space and time complexity, and find that an implementation runs $1,000\times$ faster and with $3,000\times$ less physical memory than non-sparse models. SAM learns with comparable data efficiency to existing models on a range of synthetic tasks and one-shot Omniglot character recognition, and can scale to tasks requiring 100,000s of time steps and memories. As well, we show how our approach can be adapted for models that maintain temporal associations between memories, as with the recently introduced Differentiable Neural Computer.

## 1   Introduction

Recurrent neural networks, such as the Long Short-Term Memory (LSTM) [11], have proven to be powerful sequence learning models [6, 18]. However, one limitation of the LSTM architecture is that the number of parameters grows proportionally to the square of the size of the memory, making them unsuitable for problems requiring large amounts of long-term memory. Recent approaches, such as Neural Turing Machines (NTMs) [7] and Memory Networks [21], have addressed this issue by decoupling the memory capacity from the number of model parameters. We refer to this class of models as memory augmented neural networks (MANNs). External memory allows MANNs to learn algorithmic solutions to problems that have eluded the capabilities of traditional LSTMs, and to generalize to longer sequence lengths. Nonetheless, MANNs have had limited success in real world application.

A significant difficulty in training these models results from their smooth read and write operations, which incur linear computational overhead on the number of memories stored per time step of training. Even worse, they require duplication of the entire memory at each time step to perform backpropagation through time (BPTT). To deal with sufficiently complex problems, such as processing

---

[*]These authors contributed equally.

a book, or Wikipedia, this overhead becomes prohibitive. For example, to store 64 memories, a straightforward implementation of the NTM trained over a sequence of length 100 consumes $\approx 30\,\mathrm{MiB}$ physical memory; to store 64,000 memories the overhead exceeds $29\,\mathrm{GiB}$ (see Figure 1).

In this paper, we present a MANN named SAM (sparse access memory). By thresholding memory modifications to a sparse subset, and using efficient data structures for content-based read operations, our model is optimal in space and time with respect to memory size, while retaining end-to-end gradient based optimization. To test whether the model is able to learn with this sparse approximation, we examined its performance on a selection of synthetic and natural tasks: algorithmic tasks from the NTM work [7], Babi reasoning tasks used with Memory Networks [17] and Omniglot one-shot classification [16, 12]. We also tested several of these tasks scaled to longer sequences via curriculum learning. For large external memories we observed improvements in empirical run-time and memory overhead by up to three orders magnitude over vanilla NTMs, while maintaining near-identical data efficiency and performance.

Further, in Supplementary D we demonstrate the generality of our approach by describing how to construct a sparse version of the recently published Differentiable Neural Computer [8]. This Sparse Differentiable Neural Computer (SDNC) is over $400\times$ faster than the canonical dense variant for a memory size of 2,000 slots, and achieves the best reported result in the Babi tasks without supervising the memory access.

## 2 Background

### 2.1 Attention and content-based addressing

An external memory $\mathbf{M} \in \mathbb{R}^{N \times M}$ is a collection of $N$ real-valued vectors, or *words*, of fixed size $M$. A soft *read* operation is defined to be a weighted average over memory words,

$$r = \sum_{i=1}^{N} w(i)\mathbf{M}(i)\,, \tag{1}$$

where $w \in \mathbb{R}^N$ is a vector of weights with non-negative entries that sum to one. Attending to memory is formalized as the problem of computing $w$. A *content addressable memory*, proposed in [7, 21, 2, 17], is an external memory with an addressing scheme which selects $w$ based upon the similarity of memory words to a given query $q$. Specifically, for the $i$th read weight $w(i)$ we define,

$$w(i) = \frac{f\left(d(q, \mathbf{M}(i))\right)}{\sum_{j=1}^{N} f\left(d(q, \mathbf{M}(j))\right)}, \tag{2}$$

where $d$ is a similarity measure, typically Euclidean distance or cosine similarity, and $f$ is a differentiable monotonic transformation, typically a softmax. We can think of this as an instance of kernel smoothing where the network learns to query relevant points $q$. Because the read operation (1) and content-based addressing scheme (2) are smooth, we can place them within a neural network, and train the full model using backpropagation.

### 2.2 Memory Networks

One recent architecture, Memory Networks, make use of a content addressable memory that is accessed via a series of read operations [21, 17] and has been successfully applied to a number of question answering tasks [20, 10]. In these tasks, the memory is pre-loaded using a learned embedding of the provided context, such as a paragraph of text, and then the controller, given an embedding of the question, repeatedly queries the memory by content-based reads to determine an answer.

### 2.3 Neural Turing Machine

The Neural Turing Machine is a recurrent neural network equipped with a content-addressable memory, similar to Memory Networks, but with the additional capability to write to memory over time. The memory is accessed by a controller network, typically an LSTM, and the full model is differentiable — allowing it to be trained via BPTT.

A *write* to memory,

$$\mathbf{M_t} \leftarrow (\mathbf{1} - \mathbf{R_t}) \odot \mathbf{M_{t-1}} + \mathbf{A_t} \,, \tag{3}$$

consists of a copy of the memory from the previous time step $\mathbf{M}_{t-1}$ decayed by the erase matrix $\mathbf{R}_t$ indicating obsolete or inaccurate content, and an addition of new or updated information $\mathbf{A}_t$. The erase matrix $\mathbf{R_t} = w_t^W e_t^T$ is constructed as the outer product between a set of write weights $w_t^W \in [0, 1]^N$ and erase vector $e_t \in [0, 1]^M$. The add matrix $\mathbf{A}_T = w_t^W a_t^T$ is the outer product between the write weights and a new *write word* $a_t \in \mathbb{R}^M$, which the controller outputs.

## 3   Architecture

This paper introduces *Sparse Access Memory (SAM)*, a new neural memory architecture with two innovations. Most importantly, all writes to and reads from external memory are constrained to a sparse subset of the memory words, providing similar functionality as the NTM, while allowing computational and memory efficient operation. Secondly, we introduce a sparse memory management scheme that tracks memory usage and finds unused blocks of memory for recording new information.

For a memory containing $N$ words, SAM executes a forward, backward step in $\Theta(\log N)$ time, initializes in $\Theta(N)$ space, and consumes $\Theta(1)$ space per time step. Under some reasonable assumptions, SAM is asymptotically optimal in time and space complexity (Supplementary A).

### 3.1   Read

The sparse read operation is defined to be a weighted average over a selection of words in memory:

$$\tilde{r}_t = \sum_{i=1}^{K} \tilde{w}_t^R(s_i)\mathbf{M}_t(s_i), \tag{4}$$

where $\tilde{w}_t^R \in \mathbb{R}^N$ contains $K$ number of non-zero entries with indices $s_1, s_2, \ldots, s_K$; $K$ is a small constant, independent of $N$, typically $K = 4$ or $K = 8$. We will refer to sparse analogues of weight vectors $w$ as $\tilde{w}$, and when discussing operations that are used in both the sparse and dense versions of our model use $w$.

We wish to construct $\tilde{w}_t^R$ such that $\tilde{r}_t \approx r_t$. For content-based reads where $w_t^R$ is defined by (2), an effective approach is to keep the $K$ largest non-zero entries and set the remaining entries to zero. We can compute $\tilde{w}_t^R$ naively in $\mathcal{O}(N)$ time by calculating $w_t^R$ and keeping the $K$ largest values. However, linear-time operation can be avoided. Since the $K$ largest values in $w_t^R$ correspond to the $K$ closest points to our query $q_t$, we can use an approximate nearest neighbor data-structure, described in Section 3.5, to calculate $\tilde{w}_t^R$ in $\mathcal{O}(\log N)$ time.

Sparse read can be considered a special case of the matrix-vector product defined in (1), with two key distinctions. The first is that we pass gradients for only a constant $K$ number of rows of memory per time step, versus $N$, which results in a negligible fraction of non-zero error gradient per timestep when the memory is large. The second distinction is in implementation: by using an efficient sparse matrix format such as Compressed Sparse Rows (CSR), we can compute (4) and its gradients in constant time and space (see Supplementary A).

### 3.2   Write

The write operation is SAM is an instance of (3) where the write weights $\tilde{w}_t^W$ are constrained to contain a constant number of non-zero entries. This is done by a simple scheme where the controller writes either to previously read locations, in order to update contextually relevant memories, or the *least recently accessed* location, in order to overwrite stale or unused memory slots with fresh content.

The introduction of sparsity could be achieved via other write schemes. For example, we could use a sparse content-based write scheme, where the controller chooses a query vector $q_t^W$ and applies writes to similar words in memory. This would allow for direct memory updates, but would create problems when the memory is empty (and shift further complexity to the controller). We decided upon the previously read / least recently accessed addressing scheme for simplicity and flexibility.

The write weights are defined as

$$w_t^W = \alpha_t \left( \gamma_t \, w_{t-1}^R + (1 - \gamma_t) \, \mathbb{I}_t^U \right), \tag{5}$$

where the controller outputs the interpolation gate parameter $\gamma_t$ and the write gate parameter $\alpha_t$. The write to the previously read locations $w_{t-1}^R$ is purely additive, while the least recently accessed word $\mathbb{I}_t^U$ is set to zero before being written to. When the read operation is sparse ($w_{t-1}^R$ has $K$ non-zero entries), it follows the write operation is also sparse.

We define $\mathbb{I}_t^U$ to be an indicator over words in memory, with a value of 1 when the word minimizes a usage measure $U_t$

$$\mathbb{I}_t^U(i) = \begin{cases} 1 & \text{if } U_t(i) = \min_{j=1,\ldots,N} U_t(j) \\ 0 & \text{otherwise.} \end{cases} \tag{6}$$

If there are several words that minimize $U_t$ then we choose arbitrarily between them. We tried two definitions of $U_t$. The first definition is a time-discounted sum of write weights $U_T^{(1)}(i) = \sum_{t=0}^{T} \lambda^{T-t} \left( w_t^W(i) + w_t^R(i) \right)$ where $\lambda$ is the discount factor. This usage definition is incorporated within *Dense Access Memory* (DAM), a dense-approximation to SAM that is used for experimental comparison in Section 4.

The second usage definition, used by SAM, is simply the number of time-steps since a non-negligible memory access: $U_T^{(2)}(i) = T - \max \left\{ t : w_t^W(i) + w_t^R(i) > \delta \right\}$. Here, $\delta$ is a tuning parameter that we typically choose to be 0.005. We maintain this usage statistic in constant time using a custom data-structure (described in Supplementary A). Finally we also use the least recently accessed word to calculate the erase matrix. $\mathbf{R}_t = \mathbb{I}_t^U \mathbf{1}^T$ is defined to be the expansion of this usage indicator where $\mathbf{1}$ is a vector of ones. The total cost of the write is constant in time and space for both the forwards and backwards pass, which improves on the linear space and time dense write (see Supplementary A).

### 3.3 Controller

We use a one layer LSTM for the controller throughout. At each time step, the LSTM receives a concatenation of the external input, $x_t$, the word, $r_{t-1}$ read in the previous time step. The LSTM then produces a vector, $p_t = (q_t, a_t, \alpha_t, \gamma_t)$, of read and write parameters for memory access via a linear layer. The word read from memory for the current time step, $r_t$, is then concatenated with the output of the LSTM, and this vector is fed through a linear layer to form the final output, $y_t$. The full control flow is illustrated in Supplementary Figure 6.

### 3.4 Efficient backpropagation through time

We have already demonstrated how the forward operations in SAM can be efficiently computed in $\mathcal{O}(T \log N)$ time. However, when considering space complexity of MANNs, there remains a dependence on $\mathbf{M_t}$ for the computation of the derivatives at the corresponding time step. A naive implementation requires the state of the memory to be cached at each time step, incurring a space overhead of $\mathcal{O}(NT)$, which severely limits memory size and sequence length.

Fortunately, this can be remedied. Since there are only $\mathcal{O}(1)$ words that are written at each time step, we instead track the sparse modifications made to the memory at each timestep, apply them in-place to compute $\mathbf{M_t}$ in $\mathcal{O}(1)$ time and $\mathcal{O}(T)$ space. During the backward pass, we can restore the state of $M_t$ from $M_{t+1}$ in $\mathcal{O}(1)$ time by reverting the sparse modifications applied at time step $t$. As such the memory is actually rolled back to previous states during backpropagation (Supplementary Figure 5).

At the end of the backward pass, the memory ends rolled back to the start state. If required, such as when using truncating BPTT, the final memory state can be restored by making a copy of $\mathbf{M_T}$ prior to calling backwards in $\mathcal{O}(N)$ time, or by re-applying the $T$ sparse updates in $\mathcal{O}(T)$ time.

### 3.5 Approximate nearest neighbors

When querying the memory, we can use an approximate nearest neighbor index (ANN) to search over the external memory for the $K$ nearest words. Where a linear KNN search inspects every element in

memory (taking $\mathcal{O}(N)$ time), an ANN index maintains a structure over the dataset to allow for fast inspection of nearby points in $\mathcal{O}(\log N)$ time.

In our case, the memory is still a dense tensor that the network directly operates on; however the ANN is a structured view of its contents. Both the memory and the ANN index are passed through the network and kept in sync during writes. However there are no gradients with respect to the ANN as its function is fixed.

We considered two types of ANN indexes: FLANN's randomized k-d tree implementation [15] that arranges the datapoints in an ensemble of structured (randomized k-d) trees to search for nearby points via comparison-based search, and one that uses locality sensitive hash (LSH) functions that map points into buckets with distance-preserving guarantees. We used randomized k-d trees for small word sizes and LSHs for large word sizes. For both ANN implementations, there is an $\mathcal{O}(\log N)$ cost for insertion, deletion and query. We also rebuild the ANN from scratch every $N$ insertions to ensure it does not become imbalanced.

# 4 Results

## 4.1 Speed and memory benchmarks

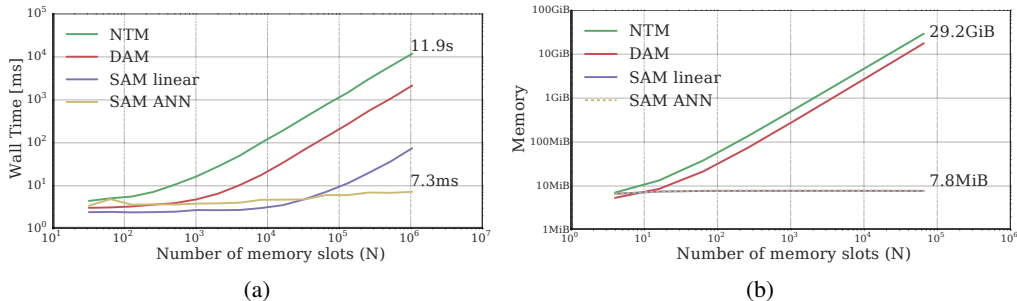

Figure 1: **(a)** Wall-clock time of a single forward and backward pass. The k-d tree is a FLANN randomized ensemble with 4 trees and 32 checks. For 1M memories a single forward and backward pass takes $12\,\mathrm{s}$ for the NTM and $7\,\mathrm{ms}$ for SAM, a speedup of $1600\times$. **(b)** Memory used to train over sequence of 100 time steps, excluding initialization of external memory. The space overhead of SAM is independent of memory size, which we see by the flat line. When the memory contains 64,000 words the NTM consumes $29\,\mathrm{GiB}$ whereas SAM consumes only $7.8\,\mathrm{MiB}$, a compression ratio of 3700.

We measured the forward and backward times of the SAM architecture versus the dense DAM variant and the original NTM (details of setup in Supplementary E). SAM is over 100 times faster than the NTM when the memory contains one million words and an exact linear-index is used, and 1600 times faster with the k-d tree (Figure 1a). With an ANN the model runs in sublinear time with respect to the memory size. SAM's memory usage per time step is independent of the number of memory words (Figure 1b), which empirically verifies the $\mathcal{O}(1)$ space claim from Supplementary A. For $64\,\mathrm{K}$ memory words SAM uses $53\,\mathrm{MiB}$ of physical memory to initialize the network and $7.8\,\mathrm{MiB}$ to run a 100 step forward and backward pass, compared with the NTM which consumes $29\,\mathrm{GiB}$.

## 4.2 Learning with sparse memory access

We have established that SAM reaps a huge computational and memory advantage of previous models, but can we really learn with SAM's sparse approximations? We investigated the learning cost of inducing sparsity, and the effect of placing an approximate nearest neighbor index within the network, by comparing SAM with its dense variant DAM and some established models, the NTM and the LSTM.

We trained each model on three of the original NTM tasks [7]. **1. Copy**: copy a random input sequence of length 1–20, **2. Associative Recall**: given 3-6 random (key, value) pairs, and subsequently a cue key, return the associated value. **3. Priority Sort**: Given 20 random keys and priority values, return

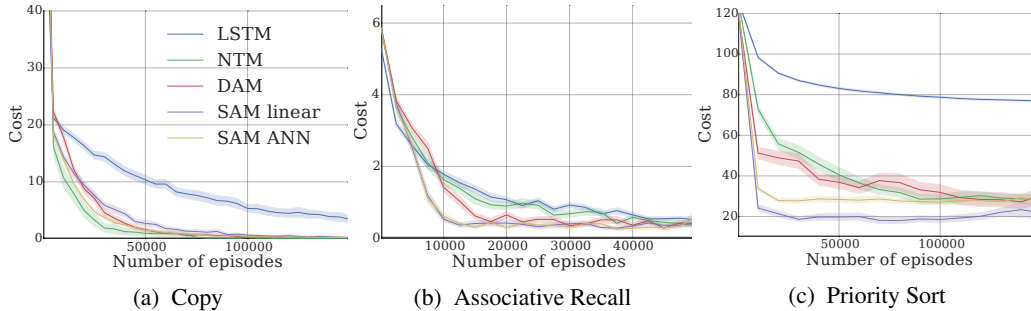

Figure 2: Training curves for sparse (SAM) and dense (DAM, NTM) models. SAM trains comparably for the Copy task, and reaches asymptotic error significantly faster for Associative Recall and Priority Sort. Light colors indicate one standard deviation over 30 random seeds.

the top 16 keys in descending order of priority. We chose these tasks because the NTM is known to perform well on them.

Figure 2 shows that sparse models are able to learn with comparable efficiency to the dense models and, surprisingly, learn more effectively for some tasks — notably priority sort and associative recall. This shows that sparse reads and writes can actually benefit early-stage learning in some cases.

Full hyperparameter details are in Supplementary C.

### 4.3 Scaling with a curriculum

The computational efficiency of SAM opens up the possibility of training on tasks that require storing a large amount of information over long sequences. Here we show this is possible in practice, by scaling tasks to a large scale via an exponentially increasing curriculum.

We parametrized three of the tasks described in Section 4.2: associative recall, copy, and priority sort, with a progressively increasing difficulty level which characterises the length of the sequence and number of entries to store in memory. For example, level specifies the input sequence length for the copy task. We exponentially increased the maximum level $h$ when the network begins to learn the fundamental algorithm. Since the time taken for a forward and backward pass scales $\mathcal{O}(T)$ with the sequence length $T$, following a standard linearly increasing curriculum could potentially take $\mathcal{O}(T^2)$, if the same amount of training was required at each step of the curriculum. Specifically, $h$ was doubled whenever the average training loss dropped below a threshold for a number of episodes. The level was sampled for each minibatch from the uniform distribution over integers $\mathcal{U}(0, h)$.

We compared the dense models, NTM and DAM, with both SAM with an exact nearest neighbor index (SAM linear) and with locality sensitive hashing (SAM ANN). The dense models contained 64 memory words, while the sparse models had $2 \times 10^6$ words. These sizes were chosen to ensure all models use approximately the same amount of physical memory when trained over 100 steps.

For all tasks, SAM was able to advance further than the other models, and in the associative recall task, SAM was able to advance through the curriculum to sequences greater than $4000$ (Figure 3). Note that we did not use truncated backpropagation, so this involved BPTT for over $4000$ steps with a memory size in the millions of words.

To investigate whether SAM was able to learn algorithmic solutions to tasks, we investigated its ability to generalize to sequences that far exceeded those observed during training. Namely we trained SAM on the associative recall task up to sequences of length $10,000$, and found it was then able to generalize to sequences of length $200,000$ (Supplementary Figure 8).

### 4.4 Question answering on the Babi tasks

[20] introduced toy tasks they considered a prerequisite to agents which can reason and understand natural language. They are synthetically generated language tasks with a vocab of about 150 words that test various aspects of simple reasoning such as deduction, induction and coreferencing.

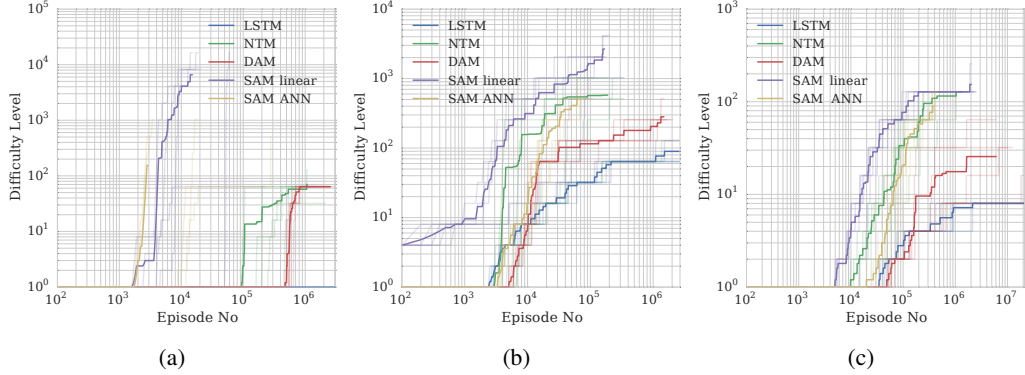

Figure 3: Curriculum training curves for sparse and dense models on (a) Associative recall, (b) Copy, and (c) Priority sort. Difficulty level indicates the task difficulty (e.g. the length of sequence for copy). We see SAM train (and backpropagate over) episodes with thousands of steps, and tasks which require thousands of words to be stored to memory. Each model is averaged across 5 replicas of identical hyper-parameters (light lines indicate individual runs).

We tested the models (including the Sparse Differentiable Neural Computer described in Supplementary D) on this task. The full results and training details are described in Supplementary G.

The MANNs, except the NTM, are able to learn solutions comparable to the previous best results, failing at only 2 of the tasks. The SDNC manages to solve all but 1 of the tasks, the best reported result on Babi that we are aware of.

Notably the best prior results have been obtained by using supervising the memory retrieval (during training the model is provided annotations which indicate which memories should be used to answer a query). More directly comparable previous work with end-to-end memory networks, which did not use supervision [17], fails at 6 of the tasks.

Both the sparse and dense perform comparably at this task, again indicating the sparse approximations do not impair learning. We believe the NTM may perform poorly since it lacks a mechanism which allows it to allocate memory effectively.

## 4.5 Learning on real world data

Finally, we demonstrate that the model is capable of learning in a non-synthetic dataset. Omniglot [12] is a dataset of 1623 characters taken from 50 different alphabets, with 20 examples of each character. This dataset is used to test rapid, or *one-shot* learning, since there are few examples of each character but many different character classes. Following [16], we generate episodes where a subset of characters are randomly selected from the dataset, rotated and stretched, and assigned a randomly chosen label. At each time step an example of one of the characters is presented, along with the correct label of the proceeding character. Each character is presented 10 times in an episode (but each presentation may be any one of the 20 examples of the character). In order to succeed at the task the model must learn to rapidly associate a novel character with the correct label, such that it can correctly classify subsequent examples of the same character class.

Again, we used an exponential curriculum, doubling the number of additional characters provided to the model whenever the cost was reduced under a threshold. After training all MANNs for the same length of time, a validation task with 500 characters was used to select the best run, and this was then tested on a test set, containing all novel characters for different sequence lengths (Figure 4). All of the MANNs were able to perform much better than chance, even on sequences $\approx 4\times$ longer than seen during training. SAM outperformed other models, presumably due to its much larger memory capacity. Previous results on the Omniglot curriculum [16] task are not identical, since we used 1-hot labels throughout and the training curriculum scaled to longer sequences, but our results with the dense models are comparable ($\approx 0.4$ errors with 100 characters), while the SAM is significantly better ($0.2 <$ errors with 100 characters).

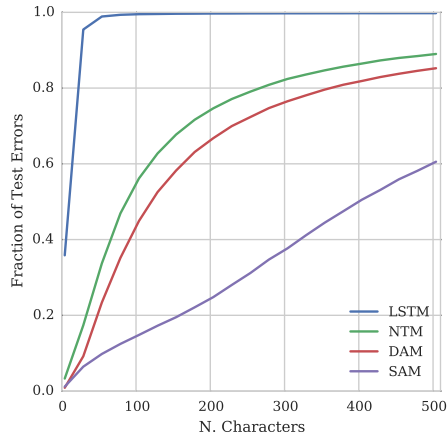

Figure 4: Test errors for the Omniglot task (described in the text) for the best runs (as chosen by the validation set). The characters used in the test set were not used in validation or training. All of the MANNs were able to perform much better than chance with $\approx 500$ characters (sequence lengths of $\approx 5000$), even though they were trained, at most, on sequences of $\approx 130$ (chance is $0.002$ for $500$ characters). This indicates they are learning generalizable solutions to the task. SAM is able to outperform other approaches, presumably because it can utilize a much larger memory.

## 5   Discussion

Scaling memory systems is a pressing research direction due to potential for compelling applications with large amounts of memory. We have demonstrated that you can train neural networks with large memories via a sparse read and write scheme that makes use of efficient data structures within the network, and obtain significant speedups during training. Although we have focused on a specific MANN (SAM), which is closely related to the NTM, the approach taken here is general and can be applied to many differentiable memory architectures, such as Memory Networks [21].

It should be noted that there are multiple possible routes toward scalable memory architectures. For example, prior work aimed at scaling Neural Turing Machines [22] used reinforcement learning to train a discrete addressing policy. This approach also touches only a sparse set of memories at each time step, but relies on higher variance estimates of the gradient during optimization. Though we can only guess at what class of memory models will become staple in machine learning systems of the future, we argue in Supplementary A that they will be no more efficient than SAM in space and time complexity if they address memories based on content.

We have experimented with randomized k-d trees and LSH within the network to reduce the forward pass of training to sublinear time, but there may be room for improvement here. K-d trees were not designed specifically for fully online scenarios, and can become imbalanced during training. Recent work in tree ensemble models, such as Mondrian forests [13], show promising results in maintaining balanced hierarchical set coverage in the online setting. An alternative approach which may be well-suited is LSH forests [3], which adaptively modifies the number of hashes used. It would be an interesting empirical investigation to more fully assess different ANN approaches in the challenging context of training a neural network.

Humans are able to retain a large, task-dependent set of memories obtained in one pass with a surprising amount of fidelity [4]. Here we have demonstrated architectures that may one day compete with humans at these kinds of tasks.

### Acknowledgements

We thank Vyacheslav Egorov, Edward Grefenstette, Malcolm Reynolds, Fumin Wang and Yori Zwols for their assistance, and the Google DeepMind family for helpful discussions and encouragement.

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
