[Supplementary Material]

## Supplementary Information

## A  Time and space complexity

Under a reasonable class of content addressable memory architectures $\mathcal{A}$, SAM is optimal in time and space complexity.

**Definition 1.** *Let $\mathcal{M}$ be a collection of real vectors $m_1, m_2, \ldots, m_N$ of fixed dimension d. Let $\mathcal{A}$ be the set of all content addressable memory data structures that store $\mathcal{M}$ and can return at least one word $m_j$ such that $D(q, m_j) \leq c\,(1 + \epsilon)$ for a given $L^p$ norm D, query vector q, and $\epsilon > 0$; provided such a memory $m_c$ exists with $D(q, m_c) = c$.*

Existing lower bounds [14, 1] assert that for any data structure $a \in \mathcal{A}$, $a$ requires $\Omega(\log N)$ time and $\Omega(N)$ space to perform a read operation. The SAM memory architecture proposed in this paper is contained within $\mathcal{A}$ as it computes the approximate nearest neighbors problem in fixed dimensions [15]. As we will show, SAM requires $\mathcal{O}(\log N)$ time to query and maintain the ANN, $\mathcal{O}(1)$ to perform all subsequent sparse read, write, and error gradient calculations. It requires $\mathcal{O}(N)$ space to initialize the memory and $\mathcal{O}(1)$ to store intermediate sparse tensors. We thus conclude it is optimal in asymptotic time and space complexity.

### A.1  Initialization

Upon initialization, SAM consumes $\mathcal{O}(N)$ space and time to instantiate the memory and the memory Jacobian. Furthermore, it requires $\mathcal{O}(N)$ time and space to initialize auxiliary data structures which index the memory, such as the approximate nearest neighbor which provides a content-structured view of the memory, and the least accessed ring, which maintains the temporal ordering in which memory words are accessed. These initializations represent an unavoidable one-off cost that does not recur per step of training, and ultimately has little effect on training speed. For the remainder of the analysis we will concentrate on the space and time cost per training step.

### A.2  Read

Recall the sparse read operation,

$$\tilde{r}_t = \sum_{i=1}^{K} \tilde{w}_t^R(s_i) \mathbf{M}_t(s_i) \ . \tag{7}$$

As $K$ is chosen to be a fixed constant, it is clear we can compute (7) in $\mathcal{O}(1)$ time. During the backward pass, we see the gradients are sparse with only $K$ non-zero terms,

$$\frac{\partial L}{\partial \tilde{w}_t^R}(i) = \begin{cases} \mathbf{M}_t(i) \cdot \frac{\partial L}{\partial \tilde{r}_t} & \text{if } i \in \{s_1, s_2, \ldots, s_K\} \\ 0 & \text{otherwise.} \end{cases}$$

and

$$\frac{\partial L}{\partial M_t}(i) = \begin{cases} \tilde{w}_t^R(i) \frac{\partial L}{\partial \tilde{r}_t} & \text{if } i \in \{s_1, s_2, \ldots, s_K\} \\ \mathbf{0} & \text{otherwise.} \end{cases}$$

where $\mathbf{0}$ is a vector of $M$ zeros. Thus they can both be computed in constant time by skipping the computation of zeros. Furthermore by using an efficient sparse matrix format to store these matrices and vectors, such as the CSR, we can represent them using at most $3K$ values. Since the read word $\tilde{r}_t$ and its respective error gradient is the size of a single word in memory ($M$ elements), the overall space complexity is $\mathcal{O}(1)$ per time step for the read.

### A.3  Write

Recall the write operation,

$$\mathbf{M}_t \leftarrow \mathbf{M}_{t-1} - \mathbf{E}_t + \mathbf{A}_t, \ , \tag{8}$$

where $\mathbf{A}_t = w_t^W a_t^T$ is the add matrix, $\mathbf{E_t} = \mathbf{M}_{t-1} \odot \mathbf{R}_t$ is the erase matrix, and $\mathbf{R}_t = \mathbb{I}_t^U \mathbf{1}^T$ is defined to be the erase weight matrix. We chose the write weights to be an interpolation between the least recently accessed location and the previously read locations,

$$w_t^W = \alpha_t \left( \gamma_t\, \tilde{w}_{t-1}^R + (1 - \gamma_t)\, \mathbb{I}_t^U \right) \ . \tag{9}$$

For sparse reads where $w_t^R = \tilde{w}_t^R$ is a sparse vector with $K$ non-zeros, the write weights $w_t^W$ is also sparse with $K + 1$ non-zeros: 1 for the least recently accessed location and $K$ for the previously read locations. Thus the sparse-dense outer product $\mathbf{A_t} = w_t^W a_t^T$ can be performed in $\mathcal{O}(1)$ time as $K$ is a fixed constant.

Since $\mathbf{R}_t = \mathbb{I}_t^U \mathbf{1}^T$ can be represented as a sparse matrix with one single non-zero, the erase matrix $\mathbf{E}_t$ can also. As $\mathbf{A_t}$ and $\mathbf{E_t}$ are sparse matrices we can then add them component-wise to the dense $\mathbf{M_{t-1}}$ in $\mathcal{O}(1)$ time. By analogous arguments the backward pass can be computed in $\mathcal{O}(1)$ time and each sparse matrix can be represented in $\mathcal{O}(1)$ space.

We avoid caching the modified memory, and thus duplicating it, by applying the write directly to the memory. To restore its prior state during the backward pass, which is crucial to gradient calculations at earlier time steps, we roll the memory it back by reverting the sparse modifications with an additional $\mathcal{O}(1)$ time overhead (Supplementary Figure 5).

The location of the least recently accessed memory can be maintained in $\mathcal{O}(1)$ time by constructing a circular linked list that tracks the indices of words in memory, and preserves a strict ordering of relative temporal access. The first element in the ring is the least recently accessed word in memory, and the last element in the ring is the most recently modified. We keep a "head" pointer to the first element in the ring. When a memory word is randomly accessed, we can push its respective index to the back of the ring in $\mathcal{O}(1)$ time by redirecting a small number of pointers. When we wish to pop the least recently accessed memory (and write to it) we move the head to the next element in the ring in $\mathcal{O}(1)$ time.

Figure 5: A schematic of the memory efficient backpropagation through time. Each circle represents an instance of the SAM core at a given time step. The grey box marks the dense memory. Each core holds a reference to the single instance of the memory, and this is represented by the solid connecting line above each core. We see during the forward pass, the memory's contents are modified sparsely, represented by the solid horizontal lines. Instead of caching the changing memory state, we store only the sparse modifications — represented by the dashed white boxes. During the backward pass, we "revert" the cached modifications to restore the memory to its prior state, which is crucial for correct gradient calculations.

## A.4 Content-based addressing

As discussed in Section 3.5 we can calculate the content-based attention, or read weights $w_t^R$, in $\mathcal{O}(\log N)$ time using an approximate nearest neighbor index that views the memory. We keep the ANN index synchronized with the memory by passing it through the network as a non-differentiable member of the network's state (so we do not pass gradients for it), and we update the index upon each write or erase to memory in $\mathcal{O}(\log N)$ time. Maintaining and querying the ANN index represents the most expensive part of the network, which is reasonable as content-based addressing is inherently expensive [14, 1].

For the backward pass computation, specifically calculating $\frac{\partial L}{\partial q_t}$ and $\frac{\partial L}{\partial \mathbf{M_t}}$ with respect to $w_t^R$, we can once again compute these using sparse matrix operations in $\mathcal{O}(1)$ time. This is because the $K$ non-zero locations have been determined during the forward pass.

Thus to conclude, SAM consumes in total $\mathcal{O}(1)$ space for both the forward and backward step during training, $\mathcal{O}(\log N)$ time per forward step, and $\mathcal{O}(1)$ per backward step.

## B  Control flow

Figure 6: Schematic showing how the controller interfaces with the external memory in our experiments. The controller (LSTM) output $h_t$ is used (through a linear projection, $p_t$) to read and write to the memory. The result of the read operation $r_t$ is combined with $h_t$ to produce output $y_t$, as well as being feed into the controller at the next timestep ($r_{t-1}$).

## C  Training details

Here we provide additional details on the training regime used for our experiments used in Figure 2.

To avoid bias in our results, we chose the learning rate that worked best for DAM (and not SAM). We tried learning rates $\{10^{-6}, 5 \times 10^{-5}, 10^{-5}, 5 \times 10^{-4}, 10^{-4}\}$ and found that DAM trained best with $10^{-5}$. We also tried values of $K$ $\{4, 8, 16\}$ and found no significant difference in performance across the values. We used 100 hidden units for the LSTM (including the controller LSTMs), a minibatch of 8, 8 asynchronous workers to speed up training, and RMSProp [19] to optimize the controller. We used 4 memory access heads and configured SAM to read from only $K = 4$ locations per head.

## D  Sparse Differentiable Neural Computer

Recently [8] proposed a novel MANN the Differentiable Neural Computer (DNC). The two innovations proposed by this model are a new approach to tracking memory freeness (dynamic memory allocation) and a mechanism for associating memories together (temporal memory linkage). We demonstrate here that the approaches enumerated in the paper can be adapted to new models by outlining a sparse version of this model, the Sparse Differentiable Neural Computer (SDNC), which learns with similar data efficiency while retaining the computational advantages of sparsity.

### D.1  Architecture

For brevity, we will only explain the sparse implementations of these two items, for the full model details refer to the original paper. The mechanism for sparse memory reads and writes was implemented identically to SAM.

It is possible to implement a scalable version of the dynamic memory allocation system of the DNC avoiding any $O(N)$ operations by using a heap. However, because it is practical to run the SDNC with many more memory words, reusing memory is less crucial so we did not implement this and used the same usage tracking as in SAM.

The temporal memory linkage in the DNC is a system for associating and recalling memory locations which were written in a temporal order, for exampling storing and retrieving a list. In the DNC this is done by maintaining a temporal linkage matrix $\mathbf{L}_t \in [0,1]^{N \times N}$. $\mathbf{L}_t[i,j]$ represents the degree to which location $i$ was written to after location $j$. This matrix is updated by tracking the precedence weighting $p_t$, where $p_t(i)$ represents the degree to which location $i$ was written to.

$$p_0 = 0 \tag{10}$$

$$p_t = (1 - \sum_i w_t^W(i))\, p_{t-1} + w_t^W \tag{11}$$

The memory linkage is updated according to the following recurrence

$$\mathbf{L}_0 = 0 \tag{12}$$

$$\mathbf{L}_t(i,j) = \begin{cases} 0 & i = j \\ (1 - w_t^W(i) - w_t^W(j))\mathbf{L}_{t-1}(i,j) + w_t^W(i)p_{t-1}(j) & i \neq j \end{cases} \tag{13}$$

$$\tag{14}$$

The temporal linkage $\mathbf{L}_t$ can be used to compute read weights following the temporal links either forward

$$f_t^r = \mathbf{L}_t w_{t-1}^r \tag{15}$$

or backward

$$b_t^r = \mathbf{L}_t^T w_{t-1}^r \tag{16}$$

The read head then uses a 3-way softmax to select between a content-based read or following the forward or backward weighting.

Naively, the link matrix requires $O(N^2)$ memory and computation although [8] proposes a method to reduce the computational cost to $O(N \log N)$ and $O(N)$ memory cost.

In order to maintain the scaling properties of the SAM, we wish to avoid any computational dependence on $N$. We do this by maintaining two sparse matrices $\mathbf{N}_t, \mathbf{P}_t \in [0,1]^{N \times \{K_L\}}$ that approximate $\mathbf{L}_t$ and $\mathbf{L}_t^T$ respectively. We store these matrices in Compressed Sparse Row format. They are defined by the following updates:

$$\mathbf{N}_0 = 0 \tag{17}$$

$$\mathbf{P}_0 = 0 \tag{18}$$

$$\mathbf{N}_t(i,j) = (1 - w_t^W(i))\, \mathbf{N}_{t-1}(i,j) + w_t^W(i)\, p_{t-1}(j) \tag{19}$$

$$\mathbf{P}_t(i,j) = (1 - w_t^W(j))\, \mathbf{P}_{t-1}(i,j) + w_t^W(j)\, p_{t-1}(i) \tag{20}$$

Additionally, $p_t$ is, as with the other weight vectors maintained as a sparse vector with at most $K_L$ non-zero entries. This means that the outer product of $w_t p_{t-1}^T$ has at most $K_L^2$ non-zero entries. In addition to the updates specified above, we also constrain each row of the matrices $\mathbf{N}_t$ and $\mathbf{P}_t$ to have at most $K_L$ non-zero entries — this constraint can be applied in $O(K_L^2)$ because at most $K_L$ rows change in the matrix.

Once these matrices are applied the read weights following the temporal links can be computed similar to before:

$$f_t^r = \mathbf{N}_t w_{t-1}^r \tag{21}$$
$$b_t^r = \mathbf{P}_t w_{t-1}^r \tag{22}$$

Note, the number of locations we read from, $K$, does not have to equal the number of outward and inward links we preserve, $K_L$. We typically choose $K_L = 8$ as this is still very fast to compute ($100\mu s$ in total to calculate $\mathbf{N}_t, \mathbf{P}_t, p_t, f_t^r, b_t^r$ on a single CPU thread) and we see no learning benefit with larger $K_L$. In order to compute the gradients, $\mathbf{N}_t$ and $\mathbf{P}_t$ need to be stored. This could be done by maintaining a sparse record of the updates applied and reversing them, similar to that performed with the memory as described in Section 3.4. However, for implementation simplicity we did not pass gradients through the temporal linkage matrices.

## D.2 Results

We benchmarked the speed and memory performance of the SDNC versus a naive DNC implementation (details of setup in Supplementary E). The results are displayed in Figure 7. Here, the computational benefits of sparsity are more pronounced due to the expensive (quadratic time and space) temporal transition table operations in the DNC. We were only able to run comparative benchmarks up to $N = 2048$, as the DNC quickly exceeded the machine's physical memory for larger values; however even at this modest memory size we see a speed increase

of $\approx 440\times$ and physical memory reduction of $\approx 240\times$. Note, unlike the SAM memory benchmark in Section 4 we plot the total memory consumption, i.e. the memory overhead of the initial start state plus the memory overhead of unrolling the core over a sequence. This is because the SDNC and DNC do not have identical start states. The sparse temporal transition matrices $\mathbf{N}_0, \mathbf{P}_0 \in [0,1]^{N \times N\{K\}}$ consume much less memory than the corresponding $\mathbf{L}_0 \in [0,1]^{N \times N}$ in the DNC.

(a)                    (b)

Figure 7: Performance benchmarks between the DNC and SDNC for small to medium memory sizes. Here the SDNC uses a linear KNN. **(a)** Wall-clock time of a single forward and backward pass. **(b)** Total memory usage (including initialization) when trained over sequence of 10 time steps.

In order to compare the models on an interesting task we ran the DNC and SDNC on the Babi task (this task is described more fully in the main text). The results are described in Supplementary G and demonstrate the SDNC is capable of learning competitively. In particular, it achieves the best report result on the Babi task.

# E   Benchmarking details

Each model contained an LSTM controller with 100 hidden units, an external memory containing $N$ slots of memory, with word size 32 and 4 access heads. For speed benchmarks, a minibatch size of 8 was used to ensure fair comparison - as many dense operations (e.g. matrix multiplication) can be batched efficiently. For memory benchmarks, the minibatch size was set to 1.

We used Torch7 [5] to implement SAM, DAM, NTM, DNC and SDNC. Eigen v3 [9] was used for the fast sparse tensor operations, using the provided CSC and CSR formats. All benchmarks were run on a Linux desktop running Ubuntu 14.04.1 with 32GiB of RAM and an Intel Xeon E5-1650 3.20GHz processor with power scaling disabled.

# F   Generalization on associative recall

Figure 8: We tested the generalization of SAM on the associative recall task. We train each model up to a difficulty level, which corresponds to the task's sequence length, of $10,000$, and evaluate on longer sequences. The SAM models (with and without ANN) are able to perform much better than chance (48 bits) on sequences of length $200,000$.

# G   Babi results

See the main text for a description of the Babi task and its relevance. Here we report the best and mean results for all of the models on this task.

The task was encoded using straightforward 1-hot word encodings for both the input and output. We trained a single model on all of the tasks, and used the 10,000 examples per task version of the training set (a small subset of which we used as a validation set for selecting the best run and hyperparameters). Previous work has reported best results (Supplementary table 1), which with only 15 runs is a noisy comparison, so we additionally report the mean and variance for all runs with the best selected hyperparameters (Supplementary table 2).

| | LSTM | DNC | SDNC | DAM | SAM | NTM | MN S | MN U |
|---|---|---|---|---|---|---|---|---|
| 1: 1 supporting fact | 28.8 | 0.0 | 0.0 | 0.0 | 0.0 | 16.4 | 0.0 | 0.0 |
| 2: 2 supporting facts | 57.3 | 3.2 | 0.6 | 0.2 | 0.2 | 56.3 | 0.0 | 1.0 |
| 3: 3 supporting facts | 53.7 | 9.5 | 0.7 | 1.3 | 0.5 | 49.0 | 0.0 | 6.8 |
| 4: 2 argument relations | 0.7 | 0.0 | 0.0 | 0.0 | 0.0 | 0.0 | 0.0 | 0.0 |
| 5: 3 argument relations | 3.5 | 1.7 | 0.3 | 0.4 | 0.7 | 2.5 | 0.3 | 6.1 |
| 6: yes/no questions | 17.6 | 0.0 | 0.0 | 0.0 | 0.0 | 9.6 | 0.0 | 0.1 |
| 7: counting | 18.5 | 5.3 | 0.2 | 0.4 | 1.9 | 12.0 | 3.3 | 6.6 |
| 8: lists/sets | 20.9 | 2.0 | 0.2 | 0.0 | 0.4 | 6.5 | 1.0 | 2.7 |
| 9: simple negation | 18.2 | 0.1 | 0.0 | 0.0 | 0.1 | 7.0 | 0.0 | 0.0 |
| 10: indefinite knowledge | 34.0 | 0.6 | 0.2 | 0.0 | 0.2 | 7.6 | 0.0 | 0.5 |
| 11: basic coreference | 9.0 | 0.0 | 0.0 | 0.0 | 0.0 | 2.5 | 0.0 | 0.0 |
| 12: conjunction | 5.5 | 0.1 | 0.1 | 0.0 | 0.1 | 4.6 | 0.0 | 0.1 |
| 13: compound coreference | 6.3 | 0.4 | 0.1 | 0.0 | 0.0 | 2.0 | 0.0 | 0.0 |
| 14: time reasoning | 56.1 | 0.2 | 0.1 | 3.8 | 4.3 | 44.2 | 0.0 | 0.0 |
| 15: basic deduction | 49.3 | 0.1 | 0.0 | 0.0 | 0.0 | 25.4 | 0.0 | 0.2 |
| 16: basic induction | 53.2 | 51.9 | 54.1 | 52.8 | 53.1 | 52.2 | 0.0 | 0.2 |
| 17: positional reasoning | 41.7 | 21.7 | 0.3 | 6.0 | 16.0 | 39.7 | 0.0 | 41.8 |
| 18: size reasoning | 8.4 | 1.8 | 0.1 | 0.3 | 1.1 | 3.6 | 24.6 | 8.0 |
| 19: path finding | 76.4 | 4.3 | 1.2 | 1.5 | 2.6 | 5.8 | 2.1 | 75.7 |
| 20: agent's motivations | 1.9 | 0.1 | 0.0 | 0.1 | 0.0 | 2.2 | 31.9 | 0.0 |
| Mean Error (%) | 28.0 | 5.2 | 2.9 | 3.3 | 4.1 | 17.5 | 3.2 | 7.5 |
| Failed tasks (err. > 5%) | 17 | 4 | 1 | 2 | 2 | 13 | 2 | 6 |

Table 1: Test results for the best run (chosen by validation set) on the Babi task. The model was trained and tested jointly on all tasks. All tasks received approximately equal training resources. Both SAM and DAM pass all but 2 of the tasks, without any supervision of their memory accesses. SDNC achieves the best reported result on this task with unsupervised memory access, solving all but 1 task. We've included comparison with memory networks, both with supervision of memories (MemNet S) and, more directly comparable with our approach, learning end-to-end (MemNets U).

| | LSTM | DNC | SDNC | DAM | SAM | NTM |
|---|---|---|---|---|---|---|
| 1: 1 supporting fact | 30.9 ± 1.5 | 2.2 ± 5.6 | 0.0 ± 0.0 | 2.9 ± 10.7 | 4.7 ± 12.8 | 31.5 ± 15.3 |
| 2: 2 supporting facts | 57.4 ± 1.2 | 23.9 ± 21.0 | 7.1 ± 14.6 | 12.1 ± 19.3 | 30.9 ± 25.1 | 57.0 ± 1.3 |
| 3: 3 supporting facts | 53.0 ± 1.4 | 29.7 ± 15.8 | 9.4 ± 16.7 | 15.3 ± 17.4 | 31.4 ± 21.6 | 49.4 ± 1.3 |
| 4: 2 argument relations | 0.7 ± 0.4 | 0.1 ± 0.1 | 0.1 ± 0.1 | 0.1 ± 0.1 | 0.2 ± 0.2 | 0.4 ± 0.3 |
| 5: 3 argument relations | 4.9 ± 0.9 | 1.3 ± 0.3 | 0.9 ± 0.3 | 1.0 ± 0.4 | 1.0 ± 0.5 | 2.7 ± 1.2 |
| 6: yes/no questions | 18.8 ± 1.0 | 2.8 ± 5.0 | 0.1 ± 0.2 | 1.9 ± 5.3 | 3.9 ± 6.7 | 18.6 ± 2.7 |
| 7: counting | 18.2 ± 1.1 | 7.3 ± 5.9 | 1.6 ± 0.9 | 4.5 ± 6.1 | 7.3 ± 6.6 | 18.7 ± 3.2 |
| 8: lists/sets | 20.9 ± 1.4 | 4.0 ± 4.1 | 0.5 ± 0.4 | 2.7 ± 5.4 | 3.6 ± 6.2 | 18.5 ± 5.9 |
| 9: simple negation | 19.4 ± 1.5 | 3.0 ± 5.2 | 0.0 ± 0.1 | 2.1 ± 5.5 | 3.8 ± 6.7 | 17.6 ± 3.4 |
| 10: indefinite knowledge | 33.0 ± 1.6 | 3.2 ± 5.9 | 0.3 ± 0.2 | 3.4 ± 8.1 | 5.7 ± 9.2 | 25.6 ± 6.9 |
| 11: basic coreference | 15.9 ± 3.3 | 0.9 ± 3.0 | 0.0 ± 0.0 | 1.5 ± 5.5 | 2.6 ± 7.9 | 15.2 ± 9.4 |
| 12: conjunction | 7.0 ± 1.3 | 1.5 ± 1.6 | 0.2 ± 0.3 | 1.8 ± 6.4 | 2.9 ± 7.9 | 14.7 ± 8.9 |
| 13: compound coreference | 9.1 ± 1.4 | 1.5 ± 2.5 | 0.1 ± 0.1 | 0.6 ± 2.2 | 1.3 ± 2.4 | 6.8 ± 3.3 |
| 14: time reasoning | 57.0 ± 1.6 | 10.6 ± 9.4 | 5.6 ± 2.9 | 11.5 ± 15.0 | 15.0 ± 12.6 | 52.6 ± 5.1 |
| 15: basic deduction | 48.1 ± 1.3 | 31.3 ± 15.6 | 3.6 ± 10.3 | 17.2 ± 19.7 | 5.5 ± 13.8 | 42.0 ± 6.9 |
| 16: basic induction | 53.8 ± 1.4 | 54.0 ± 1.9 | 53.0 ± 1.3 | 53.8 ± 1.0 | 53.6 ± 1.2 | 53.8 ± 2.1 |
| 17: positional reasoning | 40.8 ± 1.8 | 27.7 ± 9.4 | 12.4 ± 5.9 | 16.9 ± 10.3 | 20.4 ± 8.6 | 40.1 ± 1.3 |
| 18: size reasoning | 7.3 ± 1.9 | 3.5 ± 1.5 | 1.6 ± 1.1 | 1.8 ± 1.7 | 3.0 ± 1.8 | 5.0 ± 1.2 |
| 19: path finding | 74.4 ± 1.3 | 44.9 ± 29.0 | 30.8 ± 24.2 | 23.0 ± 25.4 | 33.7 ± 27.8 | 60.8 ± 24.6 |
| 20: agent's motivations | 1.7 ± 0.4 | 0.1 ± 0.2 | 0.0 ± 0.0 | 0.1 ± 0.5 | 0.0 ± 0.0 | 2.0 ± 0.3 |
| Mean Error (%) | 28.7 ± 0.5 | 12.8 ± 4.7 | 6.4 ± 2.5 | 8.7 ± 6.4 | 11.5 ± 5.9 | 26.6 ± 3.7 |
| Failed tasks (err. > 5%) | 17.1 ± 0.8 | 8.2 ± 2.5 | 4.1 ± 1.6 | 5.4 ± 3.4 | 7.1 ± 3.4 | 15.5 ± 1.7 |

Table 2: Mean and variance of test errors for the best set of hyperparameters (chosen according the validation set). Statistics are generated from 15 runs.