[Reviews · NeurIPS 2016]

Reviewer 1

Summary

The authors use fast approximate-nearest-neighbors structures and arg-top-K operations for computing the attention weights in an Neural Turing Machine. They describe experiments showing that this does not degrade training performance, and in fact allows training on larger problems.

Qualitative Assessment

The novelty in this work is thin, see e.g. section 3.3. in the memory networks paper http://arxiv.org/pdf/1410.3916.pdf; there are now several memory network papers that use memories with millions of items and hashing methods to make the lookups scalable. On the other hand, I have not seen such tools used for writeable memories, and even if there is not a conceptual leap in this paper, certainly the practical aspects of making it work are worth reporting. Nevertheless, I do recommend accepting the paper, but would like to see some changes. 1: I think it would be quite difficult to replicate the results of this paper from the descriptions in the paper. Especially the descriptions of the tasks are lacking; considering the code for the tasks from the original NTM paper have not been released, and there is no "standard" version of the tasks, it is crucial that the paper give careful descriptions of the construction of the tasks. I would also ask that the authors commit to releasing the code for their experiments. 2: Where will this fail? does it always work? The authors often discuss their model as being smooth, but note that the argmax (or K-argmax) operation is *not* smooth. With $K=1$, in the read only setting, their models are much like the MemNN-WSH in in http://arxiv.org/pdf/1503.08895.pdf (although that model does not use hashing), and that model was reported to work less well than the fully smooth model. Is it because of the $K=1$? Because of the task? I think this paper would be more valuable if there was some analysis of failure cases and some discussion of what kinds of tasks/setups allow their training to work: their model is making a discrete action, and they train without taking into consideration (as opposed to http://arxiv.org/abs/1511.07275). I think that they get results in this way is a good contribution, but it would be better to understand what are the limits. Also, I would suggest that the authors remove section 3.6, and I would also suggest the authors not call the omniglot data "non-synthetic" or "real-world".

Confidence in this Review

3-Expert (read the paper in detail, know the area, quite certain of my opinion)


Reviewer 2

Summary

Most proposed neural network models augmented with memory scale poorly in both space and time as the amount of memory grows. The authors present a memory access scheme called Sparse Access Memory, and show that it retains the representational power while training efficiently, being 1000x faster and 3000x memory efficient.

Qualitative Assessment

I do not think the paper is technically strong enough for acceptance. Efficient backpropagation and approximation nearest neighbour using sparse access memory form the main technical parts but are quite straightforward. Therefore, even though the experiments look interesting, I am not in the favour of acceptance.

Confidence in this Review

1-Less confident (might not have understood significant parts)


Reviewer 3

Summary

This paper introduced a novel memory access scheme called Sparse Access Memory (SAM) in memory augmented neural network architecture. This research area is fairly recent with the invention of Neural Turing Machines. This augment allows neural networks to learn beyond the limits of traditional LSTM model. The contribution of this paper is all writes to and reads from external memory are constrained to a fixed size, thus subset of the memory words instead of unlimited size. The authors prove the methodology and its optimality. Conceivably it will achieve good read and write performance due to reduced memory size as shown in section 4.1. But the authors also incorporate the learning costs of introducing sparsity in section 4.2 using standard NTM tasks: 1. copy. 2. Associated Call and 3. Priority sort. The combined cost are lower than NTM for some tasks (i.e. 2 and 3). I also like the fact that the authors include real-world example in section 4.4. The benchmark is using Torch7.

Qualitative Assessment

The idea is straight-forward and novel. The speedup is significant as opposed to NTM which is impressive. The fact that this sparse reads and writes by a linear or an ANN can actually benefit early-stage learning in some cases are very interesting (section 4.2). If the author could elaborate and think about how to generalize to other cases (under which cases the learning could be improved), or by accessing different ANN approaches with SAM would be very helpful for the audience. Overall it is a well written paper with all assumptions laid out clearly and architecture in the appendix. It is interesting to read.

Confidence in this Review

2-Confident (read it all; understood it all reasonably well)


Reviewer 4

Summary

This paper introduces an end-to-end differentiable memory access scheme that has the representational power of an original NTM, but is much more efficient with training wiht very large memory.

Qualitative Assessment

This paper introduces a lot of original concepts which seem to have technical brilliance. The problem it attempts to solve is well motivated and well attacked. The impact of scaling memory systems for neural networks is large, and this paper introduces the problem and solution in a clear, concise, and well-explained manner.

Confidence in this Review

1-Less confident (might not have understood significant parts)


Reviewer 5

Summary

The paper introduces Sparse Access Memory architecture for neural nets augmented with memory such as Neural Turing Machines and Memory Networks. This allows to achieve remarkable run-time and memory efficiency improvements (three orders of magnitude) compared to the existing neural nets augmented with memory. This architecture also seems to allow remarkable improvements in performance in the setting of curriculum learning and in the ability to generalize to sequences which are an order of magnitude longer than those observed in training (from length of up to 10,000 in training to sequences of length of 200,000 in testing).

Qualitative Assessment

This is potentially a groundbreaking paper. We have an actively developing field of neural networks augmented with memory starting with Neural Turing Machines and Memory Networks. So far we have observed very interesting research and experimental results in that area, but they did not look like they were at a level where they would be practically applicable. The drastic efficiency and performance improvements reported in the present paper are likely to mean that the memory augmented neural networks are now ready to be used in practice. That is likely to have far-reaching implications. The only thing I am missing in the present paper is a discussion of the reasons for drastic performance improvements in curriculum learning (and also for drastic improvements in the ability to predict sequences with length so far beyond the training range). Certainly, run-time and memory efficiency can't account for that, so something else is going on (my preliminary intuition is that it is likely that Sparse Memory Access architecture plays a similar role here to the role played by Sparse Auto-encoders in other situations; certainly, the readers would like to know the authors' thoughts on this, even if those thoughts are preliminary).

Confidence in this Review

3-Expert (read the paper in detail, know the area, quite certain of my opinion)


Reviewer 6

Summary

The paper is presenting a Neural Turting Machine (MTN) with sparse read and write access to overcome one of the major problem of MTNs that make them unsuitable for real world applications. The problem is that MTNs need prohibitively large physical memories and operations to do do Backprop, which the architecture presented in this paper hopes to solve while having comparable performance to the state of the art.

Qualitative Assessment

The paper is well structured and easy to read. If the a brief description of the memory augmeneted neural networks is presented in the background section, or an explanation of the the memory is presented to make the paper more self-contained, that would help the readers who are not experts in the field (e.g even though it is clear to those who have worked in this area, to some readers it might not be clear what the write operator ideally is supposed to do compared to conventional memories, etc.). Also the comparison in Figure 4 might not be fair. By choosing the memories so that they require the same amount of physical memory, you can compare the performance given a limited resource (which is what you have done and definitely a justifiable one!) but I think using the same memory and not the same physical memory is more informative of how the performance is. Nevertheless, it is very interesting to see how well this algortihm is performing while being so close to the theoretical bounds.

Confidence in this Review

1-Less confident (might not have understood significant parts)